# LLM-BRec: Personalizing Session-based Social Recommendation with LLM-BERT Fusion Framework

Raksha Jalan*
Raksha.Jalan@sony.com
Sony Research India
India

Tushar Prakash*
tushar121prakash@gmail.com
Sony Research India
India

Niranjan Pedenekar
Niranjan.Pedanekar@sony.com
Sony Research India
India

## ABSTRACT

Recommendation models enhance online user engagement by suggesting personalized content, boosting satisfaction and retention. Session-based Recommender systems (SR) have become a significant approach, focusing on capturing users' short-term preferences for more accurate recommendations. Recently, Session-based Social Recommendation (SSR) has emerged as a new paradigm that extends SR by incorporating users' social networks and historical sessions, aiming to offer more personalized recommendations. However, current SSR models have two significant limitations : First, they have not efficiently explored user's personalized information, as they focus only on current session information. Second, they use computationally heavy graph-based algorithms for session representations, which significantly hampers the model's efficiency, especially during inference.

To address the aforementioned problems, this paper proposes a novel fusion framework, "LLM-BRec," which incorporates Large Language Models (LLMs) and Bidirectional Encoder Representations from Transformer (BERT) to personalize SSR. Here, For session modeling, BERT's transformer architecture and self-attention mechanism are utilized to enhance its computational efficiency by emphasizing relevant contextual information. Additionally, we leverage LLM for user-profile generation to further enhance representation at inference stage. LLM-BRec has significantly reduced SSR's training and inference time and consistently outperformed the SOTA methods. Experiments on two social datasets and two non-social datasets demonstrate the effectiveness and efficiency of LLM-BRec.

## CCS CONCEPTS

• **Information systems → Information retrieval→Users and interactive retrieval→Personalization**; • **Information systems → Information retrieval → Retrieval models and ranking→ Language models**; • **Computing methodologies→Neural Networks**;

## KEYWORDS

Recommendation Systems, Social Recommendations, Large Language Models (LLMs), BERT, Session-based Recommendation

---

*Both authors contributed equally to this research

*Gen-IR@SIGIR2024, July 18, 2024, Washington DC, US*

## 1 INTRODUCTION

With the rapid growth of e-commerce and online entertainment platforms, users expect personalized items and information. The success of these platforms relies heavily on recommendation systems designed to filter irrelevant information and offer personalized recommendations. Considering sessions in Recommendation Systems is crucial for understanding the temporal dynamics of user preferences, leading to more accurate and personalized recommendations. With this intent, Session-based Recommendation(SR) exploits the commonality of objectives within close temporal proximity to predict the next user interaction within the same session.

In the early stages of SR, Anonymous Session-based Recommendations(ASR) [14] have emerged to address scenarios where user-IDs are unavailable or user-tracking is challenging, focusing on learning user preferences from sequential transition patterns in anonymous sessions. Later, Personalized Session-based Recommendations(PSR)[21] have emerged, which leverage user-IDs to enable cross-session information transfer for users with diverse preferences, thereby enhancing the effectiveness of Session-based Recommendation models.

The rising popularity of social media has transformed online spaces and digital platforms into social-centric landscapes. Popular platforms like Spotify, Instagram, and Amazon Prime enable users not only to consume content but also to share reactions with friends. Consequently, developing recommendation systems that integrate social relationships becomes crucial to adapt to the evolving dynamics of user engagement. The growing prominence of such social-centric digital landscapes, coupled with session-based recommendations (SR), has led to the development of Session-based Social Recommendation (SSR).

The general field of social recommendation has been extensively researched. However, there is limited existing literature explicitly focusing on SSR due to its relatively recent emergence. The concept of SSR was first introduced within DGRec [25], where Graph Neural Networks (GNN) are employed to aggregate neighbors' preferences for each user. Subsequently, SERec [2] proposes the utilization of heterogeneous GNNs to learn user and item representations, integrating knowledge from social networks.

Even though the aforementioned methods have demonstrated promising performance compared to traditional session-based methods [16, 35], we contend that existing session-based social recommendation methods have two significant drawbacks: (a) Existing SSR approaches have fallen short in leveraging the richness of users' personalized information, as they primarily rely on current session data and neglect exploring personalized interest trends, hindering the model's performance. For instance, SERec only concatenates user representations at the last stage of model inference, while DGRec considers only session-level interaction history, limiting

the expressiveness of user interests. (b) Recent advancements in SSR rely mostly on computationally intensive graph-based algorithms for session modeling. Though a longer training time can be acceptable, longer inference times are less tolerable due to the need for low latency. Notably, SEFrame's employment of GNNs and DGRec's Graph Attention Network(GAT)+LSTM model at session levels, significantly hamper their efficiency, especially during inference time. However, adopting lighter models for session modeling can improve model's efficiency without compromising SSR performance.

To overcome these challenges, this work proposes a novel framework, "LLM-BRec" for the Session-based Social Recommendation. LLM-BRec employs Social-aware Heterogeneous Graph(SHG) for learning enhanced user/item representations and BERT for efficient session modeling, enhancing overall performance. Additionally, It leverages the strengths of the Language Models(LLM) for user profiling.

First, a diverse knowledge graph is built to encompass universal knowledge from all interactions,item's side information and the complete social network. Recognizing the importance of leveraging social connections for modeling user preferences, this Social-aware Heterogeneous Graph is employed to learn comprehensive representations of users and items. Next for the current session, the learned user and item representations from SHG are passed to a BERT for session modeling. This aims to predict the next item with which the user will interact in the current session. Here, we devised BERT-based model, as the self-attention mechanism in BERT calculates attentions within input sequences in parallel, making it more computationally efficient and faster for learning better session representations, especially during inference.

Finally, we have found that leveraging the contextual awareness and adaptability of LLMs can serve as a powerful tool for generating user-profile. However, we emphasizes the importance of efficiently using LLMs. Given the computational expense of recurrently updating user-profiles across sessions through LLMs by incorporating current session's information and preventing data leakage during training, we advocate for generating user-profiles based on all sessions post-training and utilizing them solely during inference. This approach optimizes the utilization of LLMs' capabilities without adding computational complexity. The proposed framework is named as "LLM-BERT Fusion Framework for Recommendation(LLM-BRec)".

The key contributions of this work are summarized as follows:

- Proposed an efficient method for session modeling with BERT, which allows faster processing and reduces the computational time. Specifically, our model has reduced training time by 50% and inference time by 80% compared to SOTA SEFrame.
- First to demostrate the importance of user profiling with LLMs post-training to enhance recommendation system performance.
- For SSR, the proposed LLM-BRec has consistently outperformed SOTA models. Additionally, to show the effectiveness of LLM-BRec, we have compared model's performance on two non-social SR datasets.

## 2 RELATED WORK

In this section, we briefly cover relevant work on session- based recommendation systems, particularly session-based social recommendation models.

### 2.1 Session Based Recommendation

Session-based Recommendation (SR) can be broadly categorized into Anonymous Session-based Recommendation (ASR) and Personalized Session-based Recommendation (PSR), depending on the availability of user information. Early studies in ASR, like FPMC[24], established the foundation by focusing on sequence extraction through Markov chains. GRU4Rec [14] was the first to formalize ASR, introducing a multi-layered GRU model. NARM[16] is another notable method with attention mechanisms within GRU. Poppular approach, SR-GNN [35] represented sessions as directed subgraphs, leveraging GNNs for item transitions. Yet, these methods catered to anonymous sessions, overlooking user preferences and social network information.

Contrary to ASR, Personalized Session-based Recommendation (PSR) involves non-anonymous interactions with associated user information. This enables the linking of different sessions generated by the same user at different times. Popular model HRNN[21] proposed a hierarchical RNN model to capture users' evolving interests. A-PGNN[36] extended SR-GNN [35] for PSR and employed the attention mechanism to capture the influence of the user's past interests on the current session. PSR enables learning of users' long-term preferences and their evolution across sessions but faces challenges in incorporating inter-user influences within the dynamic context of sessions.

### 2.2 Social Recommendations

Recently, Social Recommendations have received a lot of attention since social relations give an additional perspective on user preferences along with interaction history. Early developments include MF-based approaches like SocialMF[15], TrustSVD[11], and [39] incorporated social relations with matrix factorization. The landscape of Social Recommendation evolved significantly with the emergence of deep learning technique systems[7, 9, 13, 32]. Additionally, Models such as GraphRec[8], and SocialGCN[34] emerged, incorporating GNNs and GCNs to enhance Social recommendation systems. Popular social recommendation methods, DeepSoR[32], and DiffNet++[33] have demonstrated the effectiveness of graphs and attention-based networks for capturing the user-item interaction and social relations. Later, to capture users' evolving interests, various sequential recommendation algorithms [10, 18, 25] have emerged, which extract features from social relations and user behavior sequences. The approaches outlined above are coarse-grained and struggle to capture users' dynamic preferences since they represent user-level interactions while ignoring the relevance of user profiling and temporal segmentation of interaction.

### 2.3 Session-based Social Recommendation

Session-based Social Recommendation (SSR ) aims to combine the advantages of session-based recommendation and social recommendation and provide more accurate and personalized recommendations. It predicts users' next click in the current short-term

session- based on social networks and historical sessions. The first SSR model is DGRec[25], which uses a graph attention network to model the social influence of the user However, this method is inefficient because it needs to process many additional sessions to predict the next item of the current sessions. Then, SERec [2] proposes to use a heterogeneous graph to process related users and items when making predictions for the current session. However, compared with our model in this paper, the common limitations of existing studies are: Despite their commendable efforts to integrate social networks into session-based recommendations, they still fail to take full advantage of the personalized information of the user and item when modeling the current session. Additionally, they lack potential efficiency in session modeling as they used computationally intensive graph-based methods for session modeling during inference, but lighter models can perform more efficiently without compromising SSR performance.

## 2.4 LLMs for Recommendation

Recently, there has been rising enthusiasm for using self-attention-based models like Transformer [28], GPT [22] , LLAMA [27] and BERT [6] in the representation learning since they have produced impressive results in text sequence modeling. Inspired by these advancements, Popular recommendation models like Bert4Rec[26] and Transformers4rec[4] have adopted transformers for building recommendation systems. In pursuit of enhancing the recommendation model's performance, recent efforts like [31] involve integrating pre-trained LLMs at various modeling stages. However, due to their computational cost and inefficiency for domain-specific tasks, LLMs may not always be the optimal choice. Motivated by these constraints, our objective is to illustrate the most effective approach for harnessing LLMs to enhance SSR.

## 3 PROBLEM STATEMENT

In the session-based social recommendation, we have users' interaction history $\mathcal{H}$, i.e., user-item interaction of user set $U = \{u_1, u_2, ..., u_n\}$ and item set $V = \{v_1, v_2, ..., v_m\}$ . Along with user-user social network $\mathcal{N}$. Every user, denoted as $u \in U$, is affiliated with a collection of sessions represented by $\mathcal{H}^u = \left\{ S_1^u, S_2^u, \ldots, S_{|\mathcal{H}^u|}^u \right\}$, where $S_i^u$ signifies the $i^{th}$ session of user $u$. Each session, $S_i^u$, is an ordered sequence of items the user $u$ have interacted with such as $S_i^u[t] \in V$ denoting the $t^{th}$ item in session $S_i^u$. Finally, $\mathcal{N}$ is the social network denoted as links $L = \{l_1, l_2, ...l_{|L|}\}$, where each $l$ represents a link between users $u_i$ and $u_j$, indicating a trust relationship where user $u_i$ trusts user $u_j$. The main objective of the session-based social recommendation is to predict the next item of the current session $S \notin \mathcal{H}$.

## 4 LLM-BREC FRAMEWORK

The proposed model has two main components: the Social-aware Heterogeneous Graph (SHG), which integrates knowledge from users' historical interactions $\mathcal{H}$, item category, and social network $\mathcal{N}$ to learn the enhanced user and item representations, and the BERT-based session-modeling module, which is designed to efficiently capture the temporal dynamics of user preferences in a session-aware manner, ultimately resulting in more personalised recommendations.

In the following sections, various components of the model are discussed in detail.

## 4.1 Social-aware Heterogeneous Graphs (SHG)

This component learns the global context using users' historical interaction $\mathcal{H}$ consists of all the sessions, item's side information and user social network $\mathcal{N}$. Our SHG improves upon Knowledge graph presented by [2]. Specifically, In contrast to [2], we have utilized item's side-information such as items's categories or genre to enrich the item representation from SHG $\mathcal{V}_{HG}$ during training. The impact of this enhancement is studied in section 5.5.

*4.1.1 Heterogeneous Graph.* Formally, let $\mathcal{K} = \{\mathbb{N}, \mathbb{E}\}$ represent the heterogeneous graph. The node set $\mathbb{N} = U \cup V$ encompasses all users and items involved in $\mathcal{H}$ and $\mathcal{N}$. Note that here items are already infused with side information as shown in figure 1. The edge set $\mathbb{E}$ comprises three types of directed edges: user-user edges (UU), user-item edges (UI), and item-item edges (II). Specifically, a user-user edge $(u_i, u_j) \in \mathbb{E}$ exists if user $u_i$ is followed by user $u_j$. A user-item edge $(u_i, v_j) \in \mathbb{E}$ signifies that user $u_i$ interacted with item $v_j$ in any session. Additionally, a item-item edge $(v_i, v_j) \in \mathbb{E}$ indicates a transition from item $v_i$ to $v_j$ observed in any session.

*4.1.2 Learning User/ Item Representations using Heterogeneous Graph.* In order to learn enhanced user and item representations, we have applied a Heterogeneous Graph Neural Network (HGNN)[38] to graph $\mathcal{K}$. Let's consider that the representation of a node $v$ at layer $l$ is denoted as $\mathcal{R}^l[v]$, where $v$ represents either a user or an item. The initial embedding for user/item $v$ is given by $\mathcal{R}^0[v] \in \mathbb{R}^d$, with $d$ representing the embedding size. In order to get the node representation of a node $u$ at $l^{th}$ layer from node $v$ at layer $l-1$, a message passing technique is used as both the nodes are neighbours:

$$Message_{uv}^l = W^T \cdot \mathcal{R}^{l-1}[v] + b^l \tag{1}$$

where, $W^T$ and $b^l$ are the trainable weights. To generate an updated representation for node $u$ as we move from layer $l-1$ to layer $l$, we find the significance of each connection to node $v$ within the graph $\mathcal{K}$ as the number of neighbouring users and items might be different, making the neighbouring representation less reliable.

$$att_{uv} = \phi^l \cdot \left( \sigma \left( W^T \left( \mathcal{R}^{l-1}[u] \cdot \mathcal{R}^{l-1}[v] \right) \right) + e^l \right) \tag{2}$$

where, $\sigma$ refers to sigmoid activation function and $W^T$ are the trainable weights of linear projection. $\phi \in \mathcal{R}^d$ are also trainable weights, $\mathcal{R}^{l-1}[v]$ and $\mathcal{R}^{l-1}[u]$ are the $l-1$ layer representation of node $v$ and $u$ respectively. Finally, $e^l \in \mathcal{R}^d$ is the edge feature vector of nodes $v$ and $u$. Furthermore, these attention scores are normalized for all the neighbouring nodes. Next, the contributions from all adjacent nodes are calculated as the weighted sum of all messages.

$$\widetilde{\mathcal{R}}^{l-1}[u] = \sum_{k \in \mathcal{H}_u} att_{(u,k)} . Message_{(u,k)}^{l-1} \tag{3}$$

In the final step, the updated node representation is computed using a straightforward node-specific linear transformation, followed by the activation function ReLU, which is applied to the aggregated

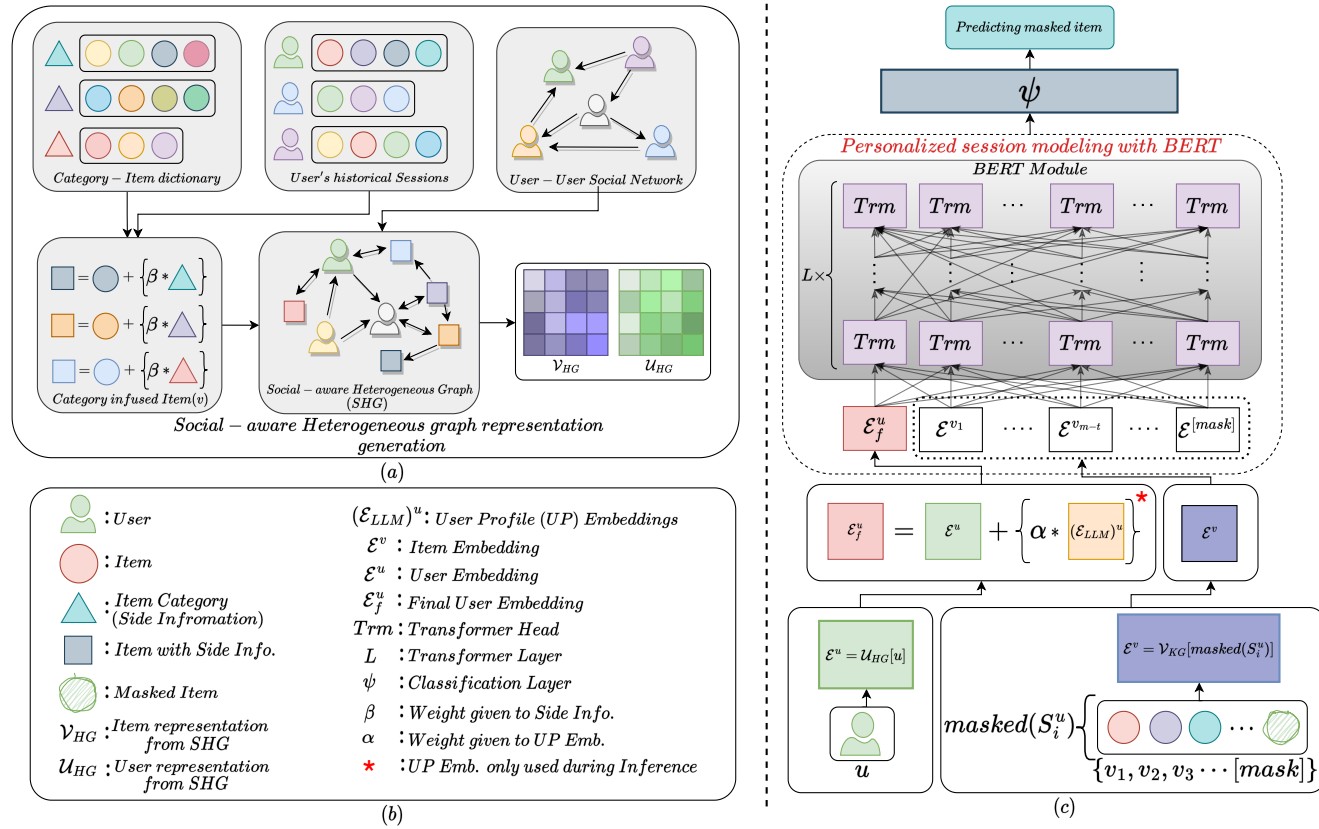

**Figure 1: (a)Social-aware Hetrogeneous Graph Module (b) List of Notations (c) The LLM-BRec Architecture**

information and the existing node representation.

$$\mathcal{R}^l[u] = ReLu(W^T(\widetilde{\mathcal{R}}^{l-1}[u]||\mathcal{R}^l[u]) + b^l) \qquad (4)$$

Where $W^T$ and $b^l$ are the trainable parameters. In this way enhanced social-aware user and item representations are learned.

## 4.2 LLMs to generate the user profile

Existing recommendation systems primarily emphasize behavior data, whereas LLMs can enhance incomplete and low-quality side information by tapping into their extensive world knowledge. When a user engages with a diverse array of items and demonstrates varied affinities across sessions, accurately recommending content solely based on user's latest session's behaviors becomes challenging.

However, using LLMs to update user-profile at each session while training is computationaly expensive as LLMs requires extensive computational resources. To address this issue, First we have tried to mimic the purpose of LLMs by getting top three categories of interacted item for each user during training. As session-based recommendation restrict access to all interactions during the training period, we dynamically identified the top three interaction categories as the session progressed. But, this strategy let to four-fold increase in training time without any significant improvement in performance.

Therefore, We propose to leverage LLMs post-training, only during inference stage. This approach for user-profiling, optimizes the utilization of LLMs' capabilities without adding computational complexity. Using LLMs for this purpose involves analyzing user interactions, such as check-in and TV viewing histories, to generate concise summaries of their preferences. These summaries, ranging from a user description of one or more sentences to a set of keywords defining user interests, are referred to as the user profile in this paper. The LLM model utilized to acquire the user profile summary is Facebook's "Llama-2-7b-chat" [27].

### 4.2.1 *Prompts for User Profiles*. To generate the user profile from LLM, two different prompt styles have been proposed.

1) **User Description $((\mathcal{UP})_1)$**: In order to get the user's interest summary in N sentences where N ∈ {1,2,3}, item attributes of a user $u$ are being passed to LLM. Let for a user $u$, $T^u = [t_1, t_2..., t_{|T^u|}]$ be the comma-separated list of item attribute of all the items that user $u$ have interacted in $\mathcal{H}_u$. These attributes might be session titles, movie titles, book titles, etc. The list is sent to the LLM with prompt 1, which requests it to summarize the user's interests in N sentences using previously interacted item attributes. The LLM outputs an N-sentence summary, identifying the user's long-term interest. Like,for Foursquare data, LLM identifies users' frequent

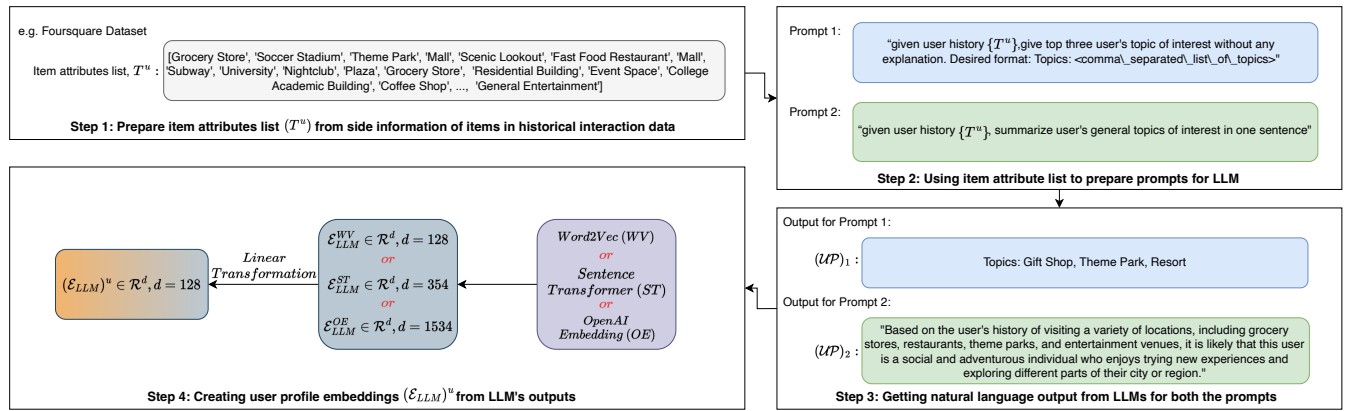

**Figure 2: Steps for Prompt and User profile Embedding Generation with Illustration**

places and characteristics, such as social or adventurous, as shown in Figure 2.

2) **User's Topic of Interest** $((\mathcal{UP})_2)$: Similarly, To get the user top $N$ topics of interest, where $N \in \{1,2,3\}$, item attribute list $T^u$ of user $u$ is passed to LLM with prompt 2. This prompt asks LLM to generate the top $N$ general topic of interest of user $u$ by analyzing the item attributes that the users have interacted with in the past. In our case, the output from LLM for this prompt is a comma-separated list containing the user's top $N$ (N=3) topic of interest, as shown in Figure 2.

*4.2.2   User profile Embedding Generation.* The user profiles generated by LLMs are in natural language and require conversion into embeddings to improve user representations. For embedding generation, we have explored the following methods:

1) **Word2Vec** ($WV$): Word2Vec[20] is a prevalent word embedding technique. The user-profile embeddings were created by averaging the pre-trained embeddings of words in LLM generated user-profile 4.2.1. This embedding, denoted as $\mathcal{E}_{LLM}^{WV}$, shares the same dimensions as the user embeddings ($\mathcal{E}^u \in \mathcal{R}^d$, where $d = 128$), eliminating the need for linear transformation.

2) **Sentence Transformer** ($ST$): The Sentence Transformer, particularly the MiniLM-L6-v2 variant[30], can understand subtleties and generate paraphrased content. Here, we have used the encoder part to get the embeddings from Sentence Transformer (MiniLM-L6-v2 variant), which is a pre-trained model.These embedding from $ST$, denoted as $\mathcal{E}_{LLM}^{ST}$ has a dimension of $d = 354$, which is adjusted using a linear projection to make it equal to the user embedding $\mathcal{E}^u \in \mathcal{R}^d$

$$(\mathcal{E}_{LLM}^{ST})^u = \sigma(W^T * \mathcal{E}_{LLM}^{ST}) \qquad (5)$$

Here, the Linear operation transforms the embedding from $\mathcal{E}_{LLM}^{ST}$ to $(\mathcal{E}_{LLM}^{ST})^u \in \mathcal{R}^d$, where the dimension $d = 128$.

3) **OpenAI Embedding Model** ($OE$): OpenAI's "text-embedding-ada-002"[17] is a sophisticated language model that excels in generating contextually rich and coherent textual representations, leveraging advanced transformer architecture and extensive pre-training.

Compared to Sentence transformer $ST$, the $OE$ model is trained on a large corpus, providing a more robust embedding representations.

These embedding from $OE$,denoted as $\mathcal{E}_{LLM}^{OE}$ has a default dimension of $d = 1534$, which is adjusted using a linear projection to make it equal to the user embedding $\mathcal{E}^u \in \mathcal{R}^d$.

$$(\mathcal{E}_{LLM}^{OE})^u = \sigma(W^T * \mathcal{E}_{LLM}^{OE}) \qquad (6)$$

Here, the Linear operation transforms the embedding from $\mathcal{E}_{LLM}^{OE}$ to $(\mathcal{E}_{LLM}^{OE})^u \in \mathcal{R}^d$, where the output dimension is $d = 128$.

These steps for user profile generation using LLMs are also presented in figure 2. More information on the pre-trained models is presented in section 5.1.3. For ease, we have referred to user profile embeddings as $\mathcal{E}_{LLM}^u$. In the next section, we will discuss how $\mathcal{E}_{LLM}^u$ embedding is used to enhance the user embedding.

## 4.3   Personalized Session-modeling with BERT

The KG embeddings encapsulate universal knowledge from all sessions and the complete social network, yet they do not encompass session-specific contextual information. Conversely, understanding user behaviors in the current session $S$ is crucial for capturing dynamic user interests. Hence, BERT[6] is employed to capture the user's present preferences and the contextual properties of items of the current session. BERT's self-attention mechanism calculates attention within input sequences in parallel, making it more computationally efficient and faster for learning better session representations.

The representations for users and items are fetched from the Social-Aware Heterogeneous Graph(SHG), denoted by $\mathcal{U}_{HG}$ and $\mathcal{V}_{HG}$ respectively. These representations serve as the user and item embeddings, which are eventually passed to BERT for session-modelling.

Specifically, for $i^{th}$ session of user $u$ denoted as $S_i^u$, the input sequence would be $S_i^u = \{v_1, v_2, ..., v_m\}$. In order to train the BERT for the next-item prediction task, the last item of the input sequence is masked. So, new item sequence will be $masked(S_i^u) = \{v_1, v_2..., [mask]\}$. Now, the embedding $\mathcal{E}^v$ of the item sequence

and user embedding $\mathcal{E}^u$ are looked-up from $\mathcal{V}_{HG}$ and $\mathcal{U}_{HG}$ respectively:

$$\mathcal{E}^v = \mathcal{V}_{HG}[masked(S_i^u)] \tag{7}$$

$$\mathcal{E}_f^u = \mathcal{U}_{HG}[u] \tag{8}$$

At training-stage, The final user embeddings is computed as mentioned in equation 8.

While at inference stage, In addition to the $\mathcal{E}_f^u$ embedding, a user profile embedding $\mathcal{E}_{LLM}^u$ generated through LLMs are also used to enhance the user representation, the process of generation of $\mathcal{E}_{LLM}^u$ embedding is present in the section 4.2. Here, $\alpha$ is the weight given to the user profile embedding $\mathcal{E}_{LLM}^u$. Therefore final user embeddings at inference would be:

$$\mathcal{E}_f^u = \mathcal{E}^u + \alpha * \mathcal{E}_{LLM}^u \tag{9}$$

Finally, in order to capture the user's evolving preferences, the user embedding $\mathcal{E}_f^u$ is concatenated with item embedding $\mathcal{E}^v$ to get a personalized representation of current session $\mathcal{E}$.

$$\mathcal{E} = \mathcal{E}_f^u \oplus \mathcal{E}^v \tag{10}$$

The embedding $\mathcal{E}$ is then passed to the BERT to predict the masked item. More details regarding the training are present in the section 4.4.

## 4.4 Model Training

The main objective of our proposed model is to predict the next item in a user's current session $u$.

Formally, Let $\theta$ be the trainable function of BERT and $\psi$ be the classification layer to predict the next item. For each iteration, embedding $\mathcal{E} \in \mathcal{R}^d, d = 128$ is generated as mentioned in section 4.3. This embedding is being passed to BERT to get the rich representation of the current session, followed by the classification layer $\psi$ to predict the masked item.

The loss function $\mathcal{L}$ of our proposed model is defined below:

$$\mathcal{L}(\mathcal{E}; \theta, \psi) = CE\_loss(\psi(\theta(\mathcal{E})), v_o) \tag{11}$$

Where $CE\_loss$ is the cross entropy loss and $v_o$ is the masked item that the model has to predict. The objective to minimise the loss function $\mathcal{L}$ is mentioned below:

$$\arg \min_{(\theta, \psi)} [\mathcal{L}(\mathcal{E}; \theta, \psi)] \tag{12}$$

The whole training process is illustrated in Figure 1 and mentioned in Algorithm 1.

At the inference-stage, we enhance user-embeddings $\mathcal{E}^u$ by adding LLM generated user-profile encodings to it.

**Table 1: Dataset Stastistical Details**

| Dataset | #users | #items | #clicks | #sessions | #links |
|---|---|---|---|---|---|
| Delicious | 1313 | 5793 | 266,190 | 60,397 | 9130 |
| Foursquare | 39,302 | 45,595 | 3,627,093 | 888,798 | 304,030 |
| ML-1M | 6034 | 3083 | 436,195 | 17005 | - |
| Amazon-book | 6136 | 7931 | 147,867 | 6841 | - |

---

**Algorithm 1** Algorithm for LLM-BRec Model Training

1: **for** each user $u$ in $U$ **do**
2:     $\mathcal{V}_{HG}, \mathcal{U}_{HG} \in \mathcal{K}$     ▷ given user and item emb. from SHG
3:     $(\mathcal{E}_{LLM})^u$     ▷ learnt user profile embedding from LLM
4:     $\mathcal{S}_i^u$     ▷ current session for user $u$
5:     $masked(\mathcal{S}_i^u) = last\_item\_masking(\mathcal{S}_i^u)$
6:     $\mathcal{E}^u = \mathcal{U}_{HG}[u]$     ▷ user emb. look-up from SHG
7:     $\mathcal{E}^v = \mathcal{V}_{HG}[mask(\mathcal{S}_i^u)]$     ▷ Item emb. look-up from SHG
8:     $\mathcal{E}_f^u = \mathcal{E}^u + \alpha * (\mathcal{E}_{LLM})^u$ ▷ $(\mathcal{E}_{LLM})^u$ used only at inference
9:     $\mathcal{E} = \mathcal{E}_f^u \oplus \mathcal{E}^v$
10:     $\hat{v} = \psi(\theta(\mathcal{E}))$
11:     $\mathcal{L} = CE\_loss(v_o, \hat{v})$     ▷ compute Cross-Entropy loss
12:     $\arg \min_{(\theta, \psi)} [\mathcal{L}(\mathcal{E}; \theta, \psi)]$
13: **end for**

---

## 5 EXPERIMENTS AND EVALUATIONS

This section aims to get answers to the following research questions:

- RQ1: How does LLM-BRec's performance compare to other state-of-the-art (SOTA) models?
- RQ2: How do variations in user-profiling method with LLM enhance the model's performance?
- RQ3: How does LLM-BRec enhance training/inference efficiency compared to SOTA models?
- RQ4: Evaluation of the quantitative impact of essential components on the overall performance of the model.

## 5.1 Experimental Settings

*5.1.1 Datasets.* We have evaluated our proposed framework using publicly available real-world datasets commonly employed in the literature for Session-based Social Recommendations (SSR):

(1) Delicious[1]: It is a collection of bookmarks from an online bookmarking system that allows users to classify bookmarks into several semantic categories. Following [2], we see a tag sequence with timestamps on a bookmark as a session, aiming to provide personalized tag suggestions. Here, we have the tag's URLs as side information, along with the session title. As mentioned in section 4.2, these session titles are used to generate user profiles with LLMs. (2) Foursquare[2]: It is a large-scale check-in dataset. The social network is built using an external social-media site. Following [2], the check-in records have been divided into sessions based on one-day intervals. Here, we have a category of checked-in places as side information.

To evaluate the adaptability of our model in non-social recommendation scenarios, we tested its performance on widely used datasets such as MovieLens-1M[3] and Amazon-Book[4] datasets. Since they originally used only interaction history, we modified our model to adapt to this use case. For user profiling in ML-1M, we used item attributes such as movie titles and genre. In Amazon Books, we utilized book titles as side information.

---

[1]https://grouplens.org/datasets/hetrec-2011/
[2]https://sites.google.com/site/yangdingqi/home/foursquare-dataset
[3]https://grouplens.org/datasets/movielens/1m/
[4]https://jmcauley.ucsd.edu/data/amazon/

For all the datasets, we used the same data partitioning and pre-processing as mentioned in [1, 2, 19, 35] to maintain consistency in results. Some statistics of the datasets after pre-processing are shown in Table 1.

*5.1.2* **Baselines**. To demonstrate the effectiveness of our proposed model, we compare its performance with various strong baselines from Session-based Recommendation(SR) and SSR.

- Anonymous Session-based Recommendations (ASR): **Item KNN** [3] model is inspired by the classic KNN model.**FPMC** [24] is a Markov-chain-based approach. **NextItNet** [37] is a CNN-based approach with dilated convolution. **NARM** [16] is an RNN-based approach that incorporates attention into GRU. **STAMP** [19] is a model that uses the attention mechanism to better understand users' short-term interests. **SR-GNN** [35] is another technique with gated GNN to learn about complex item transitions within sessions.
- Personalized Session-based Recommendations (PSR): **SSRM** [12] is the advanced SOTA approach for streaming SR.
- Session-based Social Recommendations (SSR): (1) **DGRec** [25] is the first proposed method for SSR that captures users' dynamic interests using RNNs and a graph attention network. (2) **SERec** [2] is most recent SOTA for SSR. It utilizes a heterogeneous graphs to learn embeddings and weighted GNN for current session modeling.
- **LLM-SSRM,LLM-DGRec**, **LLM-SERec**: We investigated the effectiveness of user profiling with LLMs in existing SSR. During inference, we enhanced their user embeddings with best performing LLM-based user-profile embeddings. More details in section 5.3.
- **LLM-RNNRec**, and **LLM-LSTMRec**: To investigate the impact of BERT, These are variants of our model built by replacing the BERT module with RNN and bidirectional LSTM, resp.
- **BPR** [23], **NCF** [13], **GRU4Rec** [14], **BERT4Rec** [26], **HRNN** [21]: These are popular baselines from non-social Session-based recommendation models. First two models are based on Bayesian probability and collaborative filtering respectively. GRU4Rec and HRNN use RNNs while BERT4Rec uses transformer-based architecture, BERT for current session modeling.

*5.1.3* **Experiments Details and Evaluation Metrics**. Each dataset is split into 60% training, 20% validation, and 20% testing. At the inference stage, we need to perform only the prediction task for the next session of LLM-BRec to generate predictions on the test dataset. We have adopted standard evaluation metrics Hit Ratio (HR) [5] and Mean Reciprocal Rank (MRR) [29] to evaluate the recommendation performance. To ensure consistency with SERec [2], all results are multiplied by 100. Each experiment is performed five times to eliminate inconsistency in this process.All experiments are performed on GPU:NVIDIA RTX 3090, 24GB.

For hyperparameter tuning, we used a grid search method. The number of transformer heads, transformer layers, and embedding size affected the performance. After testing, the optimal configuration was determined to be 2 transformer heads and 4 transformer layers, with an embedding size of 128 across all datasets. The user profile embedding weight, represented by $\alpha$, had the most impact

when set at 0.5, a value obtained from a range of 0.1 to 1.0. All experiments implemented early stopping based on the primary metric HR@10 to prevent overfitting.

We have used the gensim[5] library to get pre-trained Word2Vec model. Sentence Transformer Library[6] is used to get pre-trained MiniLM-L6-v2 model and OpenAI API [7] been used to get embeddings from "text-embedding-ada-002" model.

## 5.2 Performance Evaluation

The results of all experiments on the Social dataset are shown in Table 2. Our proposed model, LLM-BRec, demonstrates superior performance compared to existing baselines on both datasets.

The notable improvement of LLM-BRec over ASR, PSR, and SSR models highlights the significance of incorporating user profiling in learning better user preferences. LLM-BRec also outperforms recent PSR and SSR models, which utilize computationally heavy attention-based and graph-based models for session modeling. This signifies the effectiveness of considering the bidirectional context of user-item interactions for session-level modeling. Existing SOTA models for SSR, DGRec, and SERec perform better than non-social-aware models by leveraging social networks to learn more accurate user preferences. However, they do not outperform LLM-BRec, indicating the potency of user profiling when combined with the proposed SHG and BERT.LLM-BRec's session modeling efficiently captures contextualized sequential information, adeptly adapting to dynamic user interactions. This is in contrast to DGRec and SERec, which rely on the current session's information and complex pre-defined graph-based algorithms for session modeling.

The performance improvement of LLM-SSRM, LLM-DGRec, and LLM-SERec, i.e., adapted LLM-integrated methods over their original SOTA methods, proves the effectiveness of the proposed user profiling technique.

To validate the significance of BERT for our task, we replaced it with RNN (LLM-RNNRec) and LSTM(LLM-LSTMRec) in the proposed model. This comparison in Table 2 reveals that BERT is more capable of creating rich representations of users' session-level behavior sequences through its bi-directional context learning compared to LSTM and RNN.

In order to demonstrate the adaptability of our model for non-social SR scenarios, the results on two well-known datasets are reported in Table 3. For non-social SR systems, LLM-Brec has significantly outperformed BERT4Rec, which also uses BERT, emphasizing the importance of efficient embeddings and architecture. BPR and NCF have shown the worst performance. Our model's performance has beaten GRU4Rec and HRNN as these models utilize only current sessions and ignore the user's long-term interest trends, which we capture with user profiling.

Hence, the overall performance of LLM-BRec highlights the improvement gained by SHG for embeddings in conjunction with efficient session modeling using BERT and user profile with LLMs post-training.

---

[5]https://radimrehurek.com/gensim/models/word2vec.html
[6]https://huggingface.co/sentence-transformers/paraphrase-MiniLM-L6-v2
[7]https://platform.openai.com/docs/guides/embeddings

**Table 2: Performance Evaluation of the proposed model as compared to other methods on Delicious and Foursquare datasets. Improvement has been shown against the best-performing SOTA method, i.e., SERec( underlined). Improvements are statistically significant with p < 0.05.**

| Model | Delicious | | | | Foursquare | | | |
|---|---|---|---|---|---|---|---|---|
| | HR@10 | MRR@10 | HR@20 | MRR@20 | HR@10 | MRR@10 | HR@20 | MRR@20 |
| ItemKNN[3] | 20.84 | 9.98 | 27.82 | 10.46 | 43.88 | 23.58 | 52.11 | 24.15 |
| FPMC[24] | 29.59 | 14.46 | 38.26 | 15.02 | 44.51 | 20.93 | 55.05 | 21.66 |
| NextItNet[37] | 35.14 | 18.04 | 44.62 | 18.69 | 52.02 | 27.67 | 60.83 | 28.28 |
| NARM[16] | 37.18 | 19.76 | 46.39 | 20.40 | 53.63 | 29.40 | 62.32 | 30.00 |
| STAMP[19] | 36.29 | 19.05 | 44.96 | 19.63 | 53.12 | 28.32 | 62.14 | 29.05 |
| SR-GNN[35] | 37.01 | 19.57 | 45.74 | 20.20 | 53.19 | 28.78 | 62.07 | 29.40 |
| SSRM[12] | 37.51 | 19.83 | 46.57 | 20.46 | 53.83 | 29.33 | 62.50 | 29.93 |
| DGRec[25] | 37.78 | 20.07 | 47.36 | 20.73 | 57.05 | 31.53 | 65.85 | 32.15 |
| SERec[2] | 40.02 | 21.29 | 49.53 | 21.98 | 61.66 | 34.03 | 70.05 | 34.62 |
| LLM-SSRM | 39.12 | 21.00 | 48.49 | 21.67 | 58.13 | 32.88 | 66.96 | 33.02 |
| LLM-DGRec | 39.50 | 21.33 | 48.75 | 22.00 | 58.89 | 33.12 | 67.59 | 33.77 |
| LLM-SERec | 41.55 | 22.50 | 50.78 | 23.15 | 62.20 | 34.70 | 70.74 | 35.06 |
| LLM-RNNRec | 29.84 | 14.69 | 38.98 | 15.32 | 45.79 | 21.01 | 55.43 | 21.87 |
| LLM-LSTMRec | 35.55 | 18.63 | 44.63 | 19.26 | 53.37 | 28.11 | 61.62 | 28.79 |
| **LLM-BRec** | **42.30** | **22.92** | **51.51** | **23.55** | **63.52** | **35.93** | **71.94** | **35.98** |

**Table 3: Performance Evaluation of proposed model compared to non-social Session-based methods on MovieLens and Amazon-books datasets. Improvement has been shown against the best-performing method, i.e., HRNN (underlined). Improvements are statistically significant with p < 0.05.**

| Model | Movielens | | | | Amazon Books | | | |
|---|---|---|---|---|---|---|---|---|
| | HR@10 | MRR@10 | HR@20 | MRR@20 | HR@10 | MRR@10 | HR@20 | MRR@20 |
| BPR[23] | 5.26 | 1.83 | 8.22 | 1.97 | 45.66 | 42.83 | 46.01 | 43.24 |
| NCF[13] | 6.55 | 2.13 | 9.59 | 2.34 | 52.74 | 49.92 | 54.43 | 50.68 |
| GRU4Rec[14] | 7.27 | 2.40 | 12.12 | 2.73 | 60.26 | 58.00 | 60.72 | 58.05 |
| BERT4Rec[26] | 6.94 | 2.27 | 10.63 | 2.55 | 54.96 | 54.88 | 57.44 | 54.91 |
| HRNN[21] | 9.41 | 3.63 | 14.93 | 3.99 | 61.14 | 57.45 | 62.19 | 57.53 |
| LLM-BRec | **10.63** | **3.87** | **17.58** | **4.38** | **62.97** | **59.52** | **63.69** | **59.20** |

## 5.3 Impact of various LLM Prompts for User profiling

To examine the effects of various prompts and embedding generation techniques on user profiles created via LLMs during inference, we present two prompt variants and three embedding generation methods, as shown in Figure 2.

Multiple experiments were conducted with varying values of "N" for both prompt styles. Due to space limitations, only the performances with the optimal values are reported: Prompt 1 excels with N=3 (3 keywords), while Prompt 2 achieves optimal performance when the number of sentences is 1. Their performance is mentioned in Table 4. For this study, we've limited our data to the Delicious and Foursquare datasets, as this paper is focused on SSR. The results in Table 4 indicate that our model exhibits inferior performance when generating user profiles from Prompt 1 compared to Prompt 2. This is mainly because Prompt 2 generates a more context-rich output that naturally gives more robust user profile embedding. Here, The choice of method for user profile embedding generation significantly impacts recommendation performance. The table4 demonstrates that the user profile embedding created from

Word2Vec $((\mathcal{E}_{LLM}^{WV})^u)$ performs worse than the embedding generated from Sentence Transformer $((\mathcal{E}_{LLM}^{ST})^u)$ and OpenAI's 'text-embedding-ada-002' $((\mathcal{E}_{LLM}^{OE})^u)$. This is primarily because both the Sentence Transformer and OpenAI's model can learn enhanced representations as they are using transformer architecture. Additionally, OpenAI's method outperforms the Sentence Transformer because it's trained on a significantly larger data corpus. This naturally leads to a richer contextual representation, thus improving performance.

## 5.4 Efficiency of LLM-BRec

To demonstrate the effectiveness of our proposed model for optimizing training/inference time, we compared the models' processing times for training and inference per 1000 batches using the Foursquare dataset as presented in Table 5. The proposed framework is exceptionally efficient as the LLM-generated user-profiles and social-aware user/ item representations are pre-computed before inference. Therefore, during inference, the model just needs to process the current session with BERT architecture and predict the next interaction within the session. Table 5 clearly shows that the inference time for our LLM-BRec model is more than 80% less than

**Table 4: Performance of various user-profiling approaches**

| Dataset | Embedding | Prompt_1 | | Prompt_2 | |
|---|---|---|---|---|---|
| | | HR@10 | MRR@10 | HR@10 | MRR@10 |
| Delicious | $(\mathcal{E}_{LLM}^{WV})^u$ | 40.43 | 21.55 | 40.67 | 21.77 |
| | $(\mathcal{E}_{LLM}^{ST})^u$ | 41.05 | 21.88 | 41.75 | 22.40 |
| | $(\mathcal{E}_{LLM}^{OE})^u$ | 41.32 | 22.16 | 42.30 | 22.92 |
| Foursquare | $(\mathcal{E}_{LLM}^{WV})^u$ | 61.58 | 34.06 | 61.88 | 34.25 |
| | $(\mathcal{E}_{LLM}^{ST})^u$ | 61.75 | 34.23 | 62.40 | 35.27 |
| | $(\mathcal{E}_{LLM}^{OE})^u$ | 62.15 | 34.98 | 63.52 | 35.93 |

that of SERec. Furthermore, the training time is reduced by over 50%. This demonstrates the practicality of our model in real-world applications.

Additionally, we have performed a computational complexity analysis of our model in terms of floating-point operations (FLOPs). From Table 6, it is evident that SERec executes approximately ten times more FLOPs compared to our proposed model, LLM-BRec. As BERT processes sequences with a fixed length, resulting in a computationally efficient operation, while SERec, being graph-based, involves variable-sized graphs with higher computational overhead. This proves the superior efficiency of our model.

**Table 5: Running time in seconds per 1000 batches**

| Model | Training | Inference | Model | Training | Inference |
|---|---|---|---|---|---|
| NextItNet | 18.77 | 5.56 | SSRM | 14.63 | 5.24 |
| NARM | 11.95 | 5.08 | DGRec | 62.77 | 62.85 |
| STAMP | 11.55 | 4.98 | SERec | 54.62 | 27.52 |
| SRGNN | 27.73 | 26.61 | LLM-BRec | 24.20 | 5.02 |

**Table 6: Computational Complexity Analysis**

| Model | #FLOPs (in millions) | #Parameters (in millions) |
|---|---|---|
| SERec | 539 | 1.48 |
| LLM-BRec | 58 | 1.58 |

## 5.5 Influence of Item Side-Information on Item Representations in SHG Graphs

Table 7 illustrates the influence of item side information on learning item representations from the SHG graph $\mathcal{V}_{HG}$. Incorporating item side information enhances $\mathcal{V}_{HG}$ by capturing correlations between items. In Table 7, "SHG without side info" refers to the model where item side information isn't utilized for learning $\mathcal{V}_{HG}$. For instance, In the MovieLens dataset comprising 3083 movie IDs and 298 genres, leveraging genres enhances item representation learning by discerning correlations among diverse movies based on their genre categorizations.
Note: We've also tried adding genres as nodes in the SHG setup. These genre nodes connect with similar movies, improving how items are represented. But, doing this makes computations take longer without noticeably improving performance compared to the method mentioned earlier.

**Table 7: Impact of item's side information on item embedding**

| Model | Movielens | | Foursquare | |
|---|---|---|---|---|
| | HR@10 | MRR@10 | HR@10 | MRR@10 |
| SHG without side info. | 10.50 | 3.77 | 63.12 | 35.73 |
| LLM-BRec | 10.63 | 3.87 | 63.52 | 35.93 |

## 5.6 Component Analysis

To understand the impact of various components on the LLM-BRec model's performance, we present four model variants in Table 8. The variant **Only-SHG** model represents the current session by averaging the embeddings from the SHG for all items within the session. However, SHG lacks session-based information, resulting in the worst performance as it fails to capture the user's evolving preferences. The **Only-BERT** model utilizes the BERT module for session representation with randomly initialized user/item embeddings. Despite not incorporating SHG and user profile embeddings, this variant outperforms the previous one, highlighting the importance of efficient session modeling. The **Only-SHG+BERT** model demonstrates that combining global knowledge from SHG with efficient session modeling using BERT can lead to better performance.

To evaluate the influence of user-profiling embeddings on LLM-BRec, user profiles are fed to the model, shown as **LLM-BRec**, the model demonstrates a significant performance improvement. This highlights the capability of (LLMs) to impact the model's overall performance positively.

**Table 8: Component Analysis of LLM-BRec**

| Model | Delicious | | Foursquare | |
|---|---|---|---|---|
| | HR@10 | MRR@10 | HR@10 | MRR@10 |
| Only-SHG | 35.54 | 17.87 | 56.34 | 29.43 |
| Only-BERT | 38.82 | 20.51 | 59.64 | 32.07 |
| Only-SHG+BERT | 40.13 | 21.32 | 61.11 | 33.79 |
| LLM-BRec | 42.30 | 22.92 | 63.52 | 35.93 |

## 6 CONCLUSION

This paper introduces LLM-BRec, a fusion framework addressing limitations in Session-based Social Recommendation (SSR) systems. By utilizing LLMs for personalized user profiles and BERT for session modeling, LLM-BRec enhances recommendation performance. Experimental results on diverse datasets demonstrate its superiority over state-of-the-art methods, showcasing reduced computational costs while delivering more accurate and personalized recommendations.

LLM-BRec efficiently explores user profiling by leveraging contextual awareness from LLMs, minimizing computation by performing it only once per user post-training. BERT's transformer architecture and self-attention mechanism contribute to computational efficiency in session modeling, contrasting with graph-based algorithms. Extensive ablation studies validate the significance of each component, highlighting LLMs and BERT's roles in enhancing user interests' expressiveness and computational efficiency. Thus, LLM-BRec emerges as a promising framework for personalizing SSR systems efficiently.

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
