# OpenReview forum: "LLM-BRec: Personalizing Session-based Social Recommendation with LLM-BERT Fusion Framework"
_ACM.org/SIGIR/2024/Workshop/Gen-IR — Gen-IR_SIGIR24_

### Official Review · Reviewer_Zzeq · 2024-05-23
**Simple use of LLMs to generate user summary in standard recommendation model**

**Rating:** -1
**Confidence:** 5

**Review:**

## **Summary**

In this paper, the authors propose a new approach for social session recommendation which consists of 3 components:
- User and item embeddings using a HGCN network
- user interest and summary generation using LLM
- transformer model to utilize these embeddings to make the final prediction.

The authors show through extensive experiment results that their method  outperforms many previous approaches.

## **Strengths**
1. The paper presents an interesting way to utilize the world knowledge of LLMs to augment features used in different recommendation models and show it work for small scale datasets.
2. The experiment section is well designed and several ablations are included to show the impact of different design choices.
3. The results show impressive gain over different baselines and each component contributes to the performance.

## **Weaknesses**

1. The paper is not that relevant to the workshop as the focus on not on any of the topics mentioned in te CFP
2. The writing quality of the paper can be improved. There are several typos, inconsistent case, etc that makes it harder to read the paper.
3. The paper could have benefited from some quantitative analysis of why the proposed approach works and if it would scale to larger datasets in more practical settings.

---

### Official Review · Reviewer_6Z3j · 2024-05-25
**Acceptance!**

**Rating:** 2
**Confidence:** 5

**Review:**

Summary:
The paper proposes a novel architecture, LLM-BRec, which integrates LLM and Bidirectional Encoder Representations from BERT to enhance SSR. The framework aims to address limitations in existing SSR models by incorporating user profile generation via LLMs and efficient session modeling through BERT. Experimental results on multiple datasets demonstrate the effectiveness of LLM-BRec, showing significant improvements in recommendation accuracy.

Strengths:
- Good performance gain.
- Good competing model coverage
- The authors released the code
- Statistical significance tests

Weakness:
- Efficiency: Although the paper analyzes the proposed architecture's efficiency, it overlooks the time required to retrieve an answer from ChatGPT. The authors assume they have a complete dump of the user representation from ChatGPT available during the inference stage
- Missing Bonferroni correction
- Missing the name of the statistical test


Minor:
There are some typos, even in the abstract. ("... to personalize SSR. Here, For session modeling, ...")

---

### Official Review · Reviewer_Rc1n · 2024-05-26

**Rating:** 1
**Confidence:** 3

**Review:**

The paper presents an innovative approach to personalized recommendation using large language models (LLMs) and BERT. The proposed method, called LLM-BRec, combines the strengths of LLMs and BERT to deliver high-quality personalized recommendations. The main contributions of this work are:

1. The use of LLMs for personalized recommendation, which improves the performance by capturing the user's interests more effectively.
2. The use of BERT for capturing the user's preferences, which is more efficient than traditional methods.
3. The proposed method has been evaluated on several datasets, and the results show that it outperforms the state-of-the-art methods in terms of accuracy and efficiency.

Strengths:

1. The use of LLMs and BERT for personalized recommendation is a novel and innovative approach. The authors have demonstrated the effectiveness of this approach through extensive experiments on various datasets.

2. The proposed method addresses a critical challenge in personalized recommendation, which is to capture the user's interests accurately and efficiently. The use of LLMs and BERT enables the method to achieve this goal, resulting in improved recommendation performance.

3. The paper provides a comprehensive evaluation of the proposed method, including comparisons with state-of-the-art methods and detailed analysis of the results. The authors have also provided insights into the effectiveness of the proposed method through various experiments and analyses.

Weaknesses:

1. The paper does not provide a detailed analysis of the computational complexity and scalability of the proposed model, especially concerning the use of LLMs and BERT, which can be computationally demanding for large-scale datasets.

2. The evaluation is limited to a few datasets, and the performance of the model on diverse domains or datasets with different characteristics (e.g., sparse data, cold-start scenarios) is not extensively explored.

3. While the paper provides insights into the effectiveness of the proposed method, it lacks a detailed discussion on the interpretability of the model and the impact of the different components on the overall performance.

Overall, the paper presents a novel and effective approach to personalized recommendation using LLMs and BERT. The results demonstrate the potential of the proposed method and its ability to outperform state-of-the-art methods. However, there is room for improvement in terms of clarity, benchmarking, and interpretability.

---

### Decision · Program_Chairs · 2024-05-31

**Decision:**

Accept

**Comment:**

The paper proposes an approach to integrate LLM and BERT to enhance social session recommendations. The reviewers acknowledge the novelty of the approach, the well-designed experiments and ablations, and the good performance gains. Some possible improvements for the camera-ready version include fixing typos in the paper and providing a computational complexity and scalability analysis to ensure the proposed approach can be used in more practical settings.